# Hydroa Vacciniforme and Hydroa Vacciniforme-Like Lymphoproliferative Disorder: A Spectrum of Disease Phenotypes Associated with Ultraviolet Irradiation and Chronic Epstein–Barr Virus Infection

**DOI:** 10.3390/ijms21239314

**Published:** 2020-12-07

**Authors:** Chien-Chin Chen, Kung-Chao Chang, L Jeffrey Medeiros, Julia Yu-Yun Lee

**Affiliations:** 1Department of Pathology, Ditmanson Medical Foundation Chia-Yi Christian Hospital, Chiayi 600, Taiwan; 2Department of Cosmetic Science, Chia Nan University of Pharmacy and Science, Tainan 717, Taiwan; 3Department of Pathology, National Cheng Kung University Hospital, College of Medicine, National Cheng Kung University, Tainan 704, Taiwan; 4Department of Pathology, Kaohsiung Medical University Hospital, Kaohsiung 807, Taiwan; 5Department of Pathology, College of Medicine, Kaohsiung Medical University, Kaohsiung 807, Taiwan; 6Department of Hematopathology, The University of Texas MD Anderson Cancer Center, Houston, TX 77030, USA; ljmedeiros@mdanderson.org; 7Department of Dermatology, National Cheng Kung University Hospital, College of Medicine, National Cheng Kung University, Tainan 704, Taiwan; yylee@mail.ncku.edu.tw

**Keywords:** Epstein–Barr virus, hydroa vacciniforme, lymphoproliferative disorders, photodermatosis, sunlight, skin, ultraviolet-radiation

## Abstract

Hydroa vacciniforme (HV) is a rare form of photosensitivity disorder in children and is frequently associated with Epstein–Barr virus (EBV) infection, whereas HV-like lymphoproliferative disorders (HVLPD) describe a spectrum of EBV-associated T-cell or natural killer (NK)-cell lymphoproliferations with HV-like cutaneous manifestations, including EBV-positive HV, atypical HV, and HV-like lymphoma. Classic HV occurs in childhood with papulovesicules on sun-exposed areas, which is usually induced by sunlight and ultraviolet irradiation, and mostly resolves by early adult life. Unlike classic HV, atypical or severe HV manifests itself as recurrent papulovesicular eruptions in sun-exposed and sun-protected areas associated occasionally with facial edema, fever, lymphadenopathy, oculomucosal lesions, gastrointestinal involvement, and hepatosplenomegaly. Notably, atypical or severe HV may progress to EBV-associated systemic T-cell or natural killer (NK)-cell lymphoma after a chronic course. Although rare in the United States and Europe, atypical or severe HV and HV-like lymphoma are predominantly reported in children from Asia and Latin America with high EBV DNA levels, low numbers of NK cells, and T cell clones in the blood. In comparison with the conservative treatment used for patients with classic HV, systemic therapy such as immunomodulatory agents is recommended as the first-line therapy for patients with atypical or severe HV. This review aims to provide an integrated overview of current evidence and knowledge of HV and HVLPD to elucidate the pathophysiology, practical issues, environmental factors, and the impact of EBV infection.

## 1. Introduction

### 1.1. Hydroa Vacciniforme

Hydroa vacciniforme (HV), first described in 1862, is a rare idiopathic photosensitive cutaneous disorder [1]. Photodermatoses are a heterogeneous group of cutaneous disorders involving abnormal reactions to sunlight, usually caused or aggravated by ultraviolet (UV) components [2,3]. Initially, HV was believed to be a photodermatosis, intermediate in severity between hydroa aestivale and xeroderma pigmentosum [4]. Currently, HV is classified within the group of immunologically mediated (idiopathic) photodermatoses [2]. The prevalence of HV is 0.5 cases per 100,000 people in the Scottish population [5]. In the United States, HV constitutes 0.37% of all patients with photodermatoses and is more prevalent in non-white, non-black patients [6]. In most cases, HV occurs in childhood with a bimodal distribution of early childhood (1–7 years old) and around or after puberty (12–16 years old) [7]. Classically, HV is characterized by chronic recurrent papulovesicles or vesiculobullous eruptions on sun-exposed areas after sunlight exposure, and is usually self-limited in adolescents or young adults. Patients may be sensitive to one kind of sunlight or to a wider range of wavelengths. Some studies suggest that UVB is the causal agent [4], whereas other studies report that longer wavelengths in the UVA spectrum are the more likely cause [7,8]. HV is seasonal and occurs typically in the summer.

### 1.2. Hydroa Vacciniforme-Like Lymphoproliferative Disorders

Patients with atypical or severe HV can present with eruptions similarly to those of patients with classic HV in the early phase of the disease, but recurrent cutaneous eruptions become more severe with age, progressing to involve both sun-protected and sun-exposed areas with facial swelling, ulcers, scarring, fever, hepatitis, hematologic abnormalities, and lymphadenopathy [9]. After a protracted clinical course with many recurrences, a subset of patients with atypical or severe HV may develop a systemic and often fatal EBV-positive T cell or NK cell lymphoma, although development of lymphoma is uncommon [10]. Regarding this phenomenon, the 2008 edition of the World Health Organization (WHO) classification of hematopoietic and lymphoid tumors described the HV-like lymphoma (HVLL) mostly carrying clonal rearrangements of the T-cell receptor (TCR) genes and monoclonal Epstein–Barr virus (EBV) [11]. However, because of the inability to predict which patients will behave in an indolent fashion versus those patients who will develop overt lymphoma, the new designation hydroa vacciniforme-like lymphoproliferative disorder (HVLPD) was established in the 2016 edition of the WHO classification [12,13]. This term encompasses chronic HV-like EBV-positive lymphoproliferative disorders of childhood with an increased risk of developing systemic lymphoma. By definition, HVLPD should include EBV-positive HV subtypes, particularly atypical or severe HV cases, and HVLL with a broad spectrum of clinical severity and a long disease course. HVLPD occurs mainly in children in Asia and Latin America without a significant sex bias [14], and the median age at diagnosis is 8 years (range: 1–15 years) [13].

In the early phase of the disease, patients with HVLPD present with cutaneous disorders involving sun-exposed areas characterized by chronic papulovesicular skin lesions, ulcers, and scarring with an indolent behavior, similar to classic HV. However, some patients experience a long course of recurrences with disease extending to sun-protected skin areas, and progressing to systemic manifestations, including fever, hepatitis, lymphadenopathy, hepatosplenomegaly, and hemophagocytic syndrome [10,15,16]. Non-white patients with HVLPD are more likely to develop systemic disease than white patients [17]. Moreover, HVLPD may persist for dozens of years and over time about 15% of patients with HVLPD will develop HVLL with a higher mortality rate [10,18,19,20]. HVLL is predominantly reported in Latin America (Peru, Mexico, and Guatemala) and Asia (Korea, Japan, and Taiwan), and rarely occurs in whites [17,21]. In Taiwan, Lee et al. reported that HVLL constitutes 1% of all primary cutaneous lymphomas [22].

## 2. UV Irradiation, Chronic Ebv Infection, and the Hypothetic Pathogenesis

### 2.1. UV Irradiation May Induce HV

Among environmental hazards, UV irradiation is one of the most important factors involved in aging skin, photo-allergy, and phototoxic responses and tumor induction and progression [23]. UV irradiation can cause DNA damage, modulate the microbial landscape, and activate innate immunity by production of antimicrobial peptides and by stimulating innate cells and skin resident γδ T cells [24]. On the other hand, UV irradiation induces an immune suppressive environment in the skin, and inhibits cutaneous effector T cells, and shapes the persistence, phenotype, specificity, and function of resident memory T cells [24]. Moreover, high-dose UV irradiation may disrupt the barrier function of the skin, thereby facilitating the entrance of microbes that trigger cytokine production and modulate the immune system [24].

Since cutaneous lesions of HV patients can be induced by photoprovocation testing, the conceptually accepted hypothesis is that UV radiation with wavelengths between 320 and 390 nm may be the causal agent of HV [8,25,26,27,28,29]. A single high-dose of UVA irradiation does not induce HV, but repetitive photoprovocation tests at the same site, even with lower doses of UVA irradiation, result in the induction of erythema and vesiculation [29]. However, the mechanism and the chromophore resulting in UV-induced damage in HV are unclear and need further investigation.

### 2.2. Biological Nature of EBV and Active EBV Infection in HV

EBV belongs to the γ-herpesviridae subfamily and γ herpesviruses express virus cancer genes and immortalize infected-lymphocytes [30]. EBV has a double stranded DNA genome in its core that is 172 KB and encodes about 85 genes. Transferred via saliva in submucosal secondary lymphoid tissues, EBV has a tropism for B cells, but also can infect other types of human cells including T cells, NK cells and epithelial cells [31]. Being the most widely distributed human pathogen, EBV is associated with various diseases including benign lymphoproliferative diseases (e.g., infectious mononucleosis), lymphoid malignancies (e.g., post-transplant lymphoproliferative disease, Hodgkin lymphomas and extranodal NK/T cell lymphoma), and epithelial tumors (e.g., nasopharyngeal carcinoma and gastric carcinoma) [30]. In healthy individuals, EBV infection is kept under tight control by EBV-specific immune regulators, particularly cytotoxic CD8+ T cells which can eliminate proliferating and lytically infected B cells [32]. Reactivation of viral replication plays an important role in the development of EBV-associated malignancies [33]. During EBV-driven tumorigenesis, EBV replication during early lytic phase may facilitate an immune-suppressive tumor milieu (niche) via the chemotactic effect of CCL5 to attract monocytes, which further differentiate into tumor-associated macrophages (TAMs) [30]. Subsequently, IL-10 secretion elicited by TAMs and EBV lytic replication suppress the protective responses of cytotoxic lymphocytes, especially CD8+ T cells [30]. Furthermore, EBV-encoded microRNAs (miRNAs) attenuate the attraction of cytotoxic T lymphocytes into the tumor microenvironment through downregulation of CXCL11 expression and inhibition of MHC I-restricted antigen presentation on the tumor cells [30]. Collectively, both EBV miRNAs and early lytic replication collaborate to condition an immune suppressive microenvironment in EBV-associated cancers.

Regarding HV, EBV is highly involved in pathogenesis and frequently persists in patients [9,18,34]. In comparison with extremely low EBV DNA loads in healthy individuals, the EBV DNA load in peripheral blood mononuclear cells (PBMCs) and plasma samples is much higher in HV patients as compared with non-HV patients [35,36]. However, there is no correlation between EBV DNA loads in PBMC or plasma and HV subtype or severity [36]. EBV DNA load in PBMC or plasma also does not correlate with the survival rate of HV patients [36], but a high EBV DNA load in blood might be a useful biomarker in the diagnostic work-up [18]. In skin biopsy specimens, the number of EBV+ lymphoid cells shown by in situ hybridization for Epstein–Barr virus–encoded small RNA (EBER) is variable, although LMP-1 is usually negative (type I latency pattern). EBV-infected cells are mainly cytotoxic T-cells (CD3+, CD8 +, TIA-1+, granzyme B+, perforin+) or NK-cells (CD56+), whereas only a few are CD4+ or CD4-/CD8- [37]. Using combined flow cytometry and in situ hybridization, Kimura et al. found that EBV-infected cells in nearly half of HV or HVLL patients were γδ T cells that express many molecules characteristic of cytotoxic cells [38]. Using electron microscopy, viral particles of EBV and EBV DNA have been identified in lymphocytes and keratinocytes of phototest-induced skin lesions of HV patients, but were absent in the surrounding normal skin [18]. Notably, expression of EBV-encoded BZLF1 mRNA in skin is an EBV reactivation signal and correlates with a poorer prognosis [35,39]. Thus, although the mechanism remains unclear, EBV likely plays a key role in the pathogenesis and disease progression of HV and HVLPD.

### 2.3. Hypothetical Pathogenesis and Disease Course

Although the pathogenesis of HV and HVLPD is unclear, based on the long-term observational studies [10,18] we proposed a hypothetical course for the interplay between UV irradiation, EBV infection, HV subtypes, and HVLL (Figure 1). Cutaneous lesions of classic HV patients may be induced by UV irradiation and chronic EBV infection in childhood, usually with polyclonal T cells, and most of these lesions spontaneously remit in young adulthood. However, a subset of individuals may fail to develop immune tolerance to latent EBV infection, conferring a predisposition to acquire atypical/severe HV with UV-induced activation, and associated with atypical EBV-positive T cells. Clinically, patients with atypical/severe HV have recurrent papulovesicular eruptions and ulcerated indurated lesions on sun-exposed areas, and extending to sun-protected areas with systemic symptoms. Although most patients with atypical or severe HV have an indolent behavior with a long clinical course, in a few patients with atypical or severe HV these lesions evolve into aggressive EBV-positive HVLL, usually associated with monoclonal EBV-infected T cells. With prolonged clinical follow up, the mortality rate for patients with EBV-positive HVLL is high.

## 3. Clinical Manifestations and Laboratory Characteristics

The classic presentation of HV is that patients develop pruritus and a stinging sensation followed by hemorrhagic, umbilicated papulovesicular lesions and crusted bullae primarily on light-exposed areas [6]. Cutaneous lesions usually occur hours to days after sunlight exposure and are located mostly on the face, neck, forearms, and the dorsa of the hands [6]. The papulovesicular lesions or bullae most often develop an erythematous base and are initially tense, followed by progressive necrosis that leads to healing with a varioliform scar (Figure 2). In addition, a mixture of hyper- and hypopigmented lesions results in a polymorphous skin presentation. The differential diagnosis for HV at this stage includes erythropoietic and hepatic porphyrias, which can be excluded by checking the level of porphyrins in the blood and urine, as well as phototoxic reactions, a vesiculobullous form of polymorphous light eruption, and actinic prurigo [40]. Classic HV has no systemic symptoms and no abnormalities in routine laboratory tests, immune functions, anti-nuclear factors, vitamin metabolism, or porphyrin levels [9]. Classic HV usually remits during adolescence and is not associated with systemic disease or fatal morbidity.

Unlike classic HV, patients with atypical or severe HV may have skin lesions on sun-protected areas, and can present with systemic symptoms such as fever, lymphadenopathy, hepatosplenomegaly, anemia, and leukopenia, particularly late in the course of disease (Figure 3). The cutaneous signs of atypical or severe HV occur in patients at an older age than that of patients with classic HV, and age of onset >9 years is a risk factor [35]. In one study, conjunctivitis and oral aphthous stomatitis/gingivitis were observed in 26% of patients with classic HV versus 42% of patients with atypical/severe HV [35]. In addition, over 90% of patients with classic HV or atypical/severe HV have elevated γδ T-cell percentages > 5%, and all HV patients have a high EBV DNA load, although there were no statistically significant differences between classic versus atypical/severe HV [35]. By real-time quantitative reverse transcription polymerase chain reaction (qRT-PCR) and flow cytometry immunophenotypic analysis, Cohen et al. described elevated levels of EBV DNA in the blood and high levels of EBV DNA in T or NK cells in all HVLPD patients, without significant differences between patients with classic versus atypical/severe HV [17]. Compared to classic HV patients, atypical/severe HV patients more frequently had abnormal antibody profiles to EBV antigens, such as elevated antiviral capsid antigen (VCA), elevated anti-early antigen (EA) and low EBV nuclear antigen (EBNA), and the serologic patterns were consistent with chronic active EBV infection [9]. However, anti-EBV antibody titres, EBV DNA load in PBMCs, number of EBER positive cells, and the subsets of infiltrating cells in skin biopsy specimens did not correlate with mortality [35].

When extracutaneous involvement occurs (e.g., hepatosplenomegaly, lymphadenopathy, and bone marrow infiltration) with hematological abnormalities, Gru and Jaffe recommend that these lesions be classified as HVLL [37]. Fever, weight loss, and asthenia can be present, and elevated serum LDH and/or liver function tests are noted in one-third of patients [37]. Clonality testing can be helpful because the presence of monoclonal T-cell receptor gene rearrangements correlates with progression of HVLPD to HVLL.

## 4. Phototesting and Photoprovocation Testing

Phototesting is an important method to confirm photosensitivity and provide an action spectrum, although the methods employed for phototesting vary in different dermatology units. Generally, using a template with several exposable windows, unaffected skin (preferably on the back or abdomen) is exposed to different doses of UVA, broadband UVB, and/or visible monochromatic or broad-spectrum radiation. The assessment consists of immediate observation for possible urticarial lesions (solar urticaria) and the minimal erythema dose (MED) reading performed 24 h after the exposures [2,41]. The MED is defined as the dose of UVB, UVA, or visible light that induces perceptible erythema on the irradiated area [41]. At least 2 weeks prior phototesting, systemic immunosuppressants and topical agents should be halted to avoid possible interference [2]. In patients with HV, with a source of polychromatic UVA, the UVA MED can be low [26]. With a source of monochromatic UVA, Sonnex et al. observed low MEDs between 340 and 380 nm in three patients [42], but others demonstrated normal UVA MEDs in HV patients [10,43].

Photopatch testing is diagnostically useful to evaluate patients with photoallergic contact dermatitis, and a positive reaction involving unirradiated and irradiated sites is consistent with allergic contact dermatitis. Photopatch testing is usually negative in patients with HV [41]. Photo-provocation testing is employed to induce photosensitivity and can be performed by exposing the same skin site to three to five consecutive days of exposure. In about half of patients with HV, lesions can be reproduced by photoprovocative testing [18]. Lesions that develop after photoprovocation can be assessed clinically and biopsied for histopathologic evaluation.

## 5. Pathological Features and Immunochemical Profiles

Histopathological features of HV can vary according to clinical stage and the severity of different lesions. In early stages of HV, characteristic histologic features in the skin include intraepidermal spongiotic vesiculation, varying degrees of lymphocytic infiltrate in the upper dermis (Figure 2) and areas of keratinocyte necrosis [7,10]. A histologic hallmark of HV, suggested by Iwatsuki et al., is a dense perivascular lymphocytic infiltration with reticulated degeneration of the epidermis [9]. The histologic features of classic HV and atypical/severe HV are essentially indistinguishable, such as reticulated epidermal degeneration and dense perivascular infiltration of mainly T cells, although the lymphoid infiltrate is often more dense and extensive, and extends deeply into subcutaneous tissue in atypical/severe HV [9]. Moreover, the clinical manifestations often deviate from the pathological features. Even with clinical features of aggressive disease or systemic involvement, lymphoid cells often show minimal or mild atypia (Figure 3) [44,45,46]. Thus, the diagnosis of HV versus HVLPD relies greatly on a careful history, physical examination, and phototesting, whereas histopathologic examination and laboratory tests are helpful in ruling out other photodermatoses.

Notably, in situ hybridization for EBER in skin biopsy specimens shows positive cells in over 95% of patients with classic HV and atypical/severe HV [9] and 100% patients with HVLPD [19]. Immunohistochemically, expression of CD5, CD7, CD43, and CD25 is variable, whereas CD57 is negative [37]. The lymphoid infiltrate is composed most frequently of cytotoxic CD8+ T cells, but in one-third of cases the infiltrate is derived from CD56+ NK cells [46]. The T cells can be positive for either T-cell receptor αβ or γδ. The Ki-67 index is variable, and can be very low or as high as 50% [37].

Monoclonal rearrangement of *TRG* and/or *TRB* is regarded as a useful tool for distinguishing classic HV from severe HV [29]. In a large cohort study of HVLPD, 88% of patients carried monoclonal T-cell receptor gene rearrangements [19]. Xie et al. further recommended that clonal rearrangements of the T-cell receptor genes is a prognostic indicator. In their study all patients with monoclonal T-cell receptor gene rearrangements died [45]. In our previous study, all classic HV patients showed no evidence of a monoclonal T-cell population in the initial skin specimens [10]. Thus, T-cell monoclonality would be unusual in classic HV and might be an important clue to herald progression to atypical HV or HVLL, especially combined with an atypical or aggressive clinical presentation [10,20]. Notably, a subset of HV-like eruptions is one of the clinical manifestations of chronic active EBV infection (CAEBV), and clinicopathologic survey with fulfillment of diagnostic criteria is the key to reach the diagnosis of CAEBV [13]. Direct immunofluorescence is usually nonspecific, although there are a few reports describing granular deposition of C3 below the basement membrane and in the dermal papillae [8,42].

## 6. Molecular and Genetic Characteristics

Given the rare occurrence of HVLPDs, the reports regarding molecular and genetic features of these lesions are limited. Since Zhang et al. reported chromosome 6q deletion in a cell line (SNK-11) from a patient with HV [47], some genomic breakthroughs have been explored. In 2016, Cohen et al. reported that GATA2 deficiency is associated with EBV-positive HVLL [48]. GATA2 is a transcription factor crucial for both cellular immune responses and controlling the latent infection of herpesviruses, and is expressed in hematopoietic progenitors. Because one of GATA2 binding sites in EBV is located in a latency promoter, the Cp promoter for EBNA latency proteins, insufficiency of GATA2 may impair virus latency and result in more viral replication with prolonged active EBV infection and low numbers of monocytes, CD4 T cells, B cells, and NK cells [48]. To date, only one 24-year-old Cantonese woman with HVLL was proven to have GATA2 deficiency and was successfully treated with a haplo-identical hematopoietic cell transplant from her unaffected sister [48]. Except the aforementioned case, Cohen et al. found no mutations in GATA2 or other genes associated with severe EBV disease or repair of DNA in HVLPD patients [17].

Cohen and colleagues performed RNAseq analysis on a skin specimen of one patient with HVLPD. Compared with normal control tissue, skin of HVLPD revealed upregulation of genes encoding multiple chemokines [17], with the most upregulated genes being CXCL11, CXCL9, CXCL10, and CCL4. These genes encode chemoattractants for activated T cells, activated monocytes, and NK cells. Other upregulated genes include *IFNG*, which encodes interferon γ to inhibit EBV outgrowth, and *APOBEC3A*, which encodes a member of the cytidine deaminase family to suppress replication of cytomegalovirus [17].

Moreover, Cohen et al. conducted a whole-exome sequencing study on a Caucasian patient with HVLPD and did not identify any pathogenic variants [17]. However, by performing a whole-exome sequencing study in five Chinese patients with HVLPD, Xie et al. reported eight mutant genes that might correlate with development of HVLPD [45]. The eight mutant genes include five driver mutations involving *STAT3, IKBKB, ELF3, CHD7*, and *KMT2D*, as well as three passenger mutations involving *ELK1, RARB*, and *HPGDS*. All of these mutant genes and their downstream signaling pathways have been shown to be pathogenic factors in lymphoproliferative diseases [45]. The occurrence of these mutant genes and the potential mechanisms employed by these genes in HVLPD warrants further study.

## 7. Treatment and Therapeutic Agents

Traditionally, HV is treated conservatively, since classic HV usually undergoes spontaneous remission before adulthood. The therapeutic strategies consist of broad-spectrum sunscreens with a high sun protection factor and avoidance of sunlight exposure [7,28,40]. In patients who do not respond to conservative treatment or have a prolonged refractory course, dermatologists should consider the possibility of HVLPD, particularly atypical/severe HV or HVLL. Currently, there are no guidelines to treat patients with HVLPD, mostly owing to the rarity of the disease. Several regimens, including phototherapy [41], immunomodulatory agents (antiviral agents, interferon, thalidomide, hydroxychloroquine, intravenous immunoglobulin, etc.) [17,19,49,50,51,52], systemic corticosteroids [17,19], chemotherapy [19,53], and hematopoietic stem cell transplantation [19,53] have been used with variable rates of success. Regarding phototherapy, narrowband UVB (NBUVB) phototherapy three times weekly for five weeks is most often recommended [41], whereas psoralen-UVA (PUVA) photochemotherapy is not recommended for younger patients. In a report of 41 cases with HVLPD, Liu et al. suggested conservative treatment for most Chinese patients with HVLPD, and that chemotherapy should not be used as a first-line treatment [19]. Similarly, studies from Latin American nations have reported poor outcomes in patients receiving chemotherapy [54]. If chemotherapy fails to eradicate lymphoma cells, this therapy may worsen the prognosis by inducing immunosuppression and reactivating EBV replication. Among a variety of therapeutic modalities, immunomodulatory therapy for HVLPD is often recommended [37]. Immunomodulator agents can modulate the host inflammatory response by enhancing innate immunity against viral replication, and can suppress disease aggressiveness.

## 8. Conclusions

HV (primarily cutaneous manifestations) and HVLPD (more often systemic or extracutaneous involvement) encompass a range of disease entities that typically occur in children or young adults who usually have a prolonged clinical course. HV and HVLPD are highly associated with EBV infection and UV radiation. Despite the clinical and pathological similarities between HV and HVLPD, prognosis is divergent (Table 1). Establishing a diagnosis of HV, HVLPD, or HVLL is multifactorial and therefore needs to incorporate history, clinical manifestations, phototesting results, histopathological features, presence and extent of EBV infection, and assessment of T-cell clonality. Because patients with HVLPD exhibit a highly variable clinical course, close monitoring and systemic evaluation are essential to alleviate the potential risk of systemic lymphoma. Although there is no consensus on a first-line therapy for patients with HVLPD, immunomodulatory therapies may be associated with greater efficacy than chemotherapy. Molecular testing of HV and HVLPD is needed to identify biomarkers helpful in predicting clinical behavior, monitoring disease activity, and for identifying potential targets for drug development, thereby facilitating personalized treatment of patients with HVLPD. A major challenge in these efforts is the rarity of this disease, and therefore it seems likely that targeted funding and large cooperative efforts between multiple institutions will be required to more thoroughly study this disease.

## Figures and Tables

**Figure 1 ijms-21-09314-f001:**
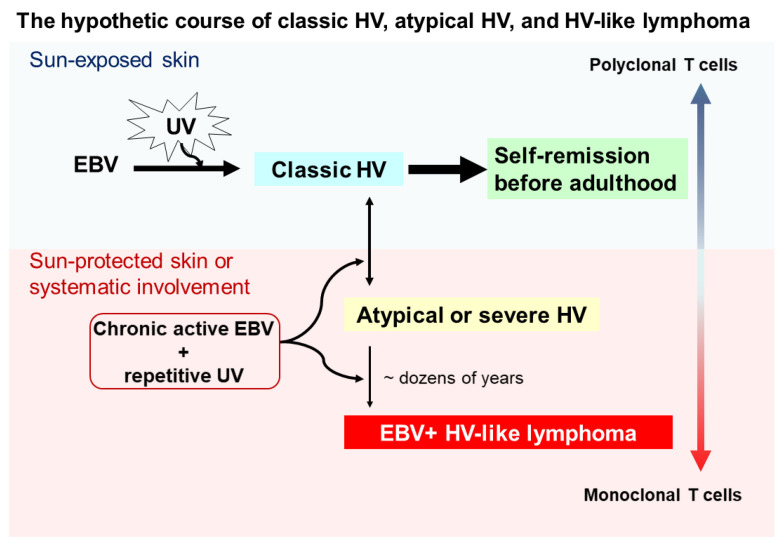
Schematic of a hypothetical suggestion regarding the onset and evolution of classic hydroa vacciniforme (HV), atypical or severe HV, and hydroa vacciniforme-like lymphoma (HVLL). Ultraviolet (UV) irradiation and chronic Epstein–Barr virus (EBV) infection likely contribute to the onset of classic HV skin lesions in children, but most of these lesions resolve spontaneously by young adulthood. However, under chronic active EBV infection and repetitive UV exposure, some patients develop atypical or severe HV characterized by recurrent papulovesicular eruptions and ulcerated indurated lesions on sun-exposed areas. These lesions also may extend to sun-protected areas with systemic symptoms (e.g., fever), liver damage, lymphadenopathy, hepatosplenomegaly, and hemophagocytic syndrome. Despite most patients with atypical or severe HV having an indolent clinical course, a few patients with atypical or severe HV after dozens of years evolve into aggressive EBV-positive HVLL, associated with a monoclonal T-cell proliferation.

**Figure 2 ijms-21-09314-f002:**
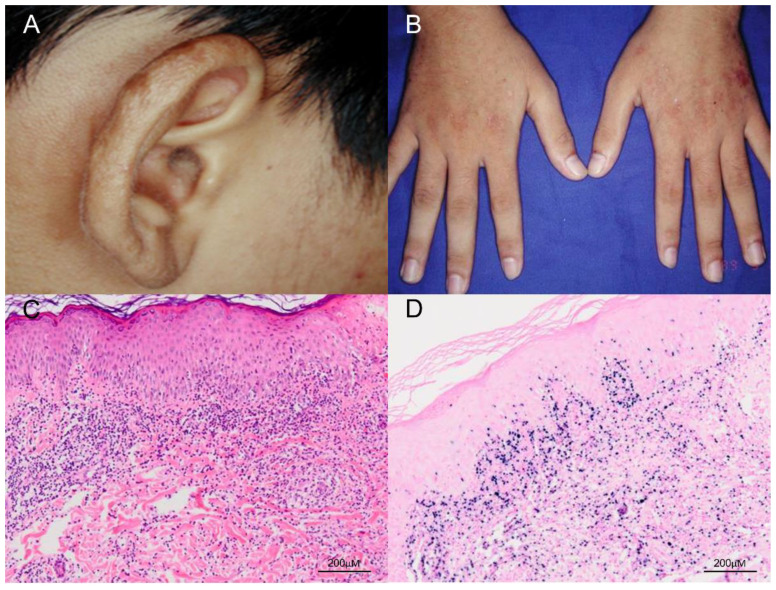
Clinical and pathologic features of classic HV. (**A**,**B**) The patient was a 17-year-old boy who presented with multiple papulovesicular eruptions, crusting and scarring over the earlobes, face and the dorsum of both hands for 4–5 years. The lesions were exacerbated in summer. (**C**) Histological examination of the skin biopsy specimen shows a dense lymphocytic infiltrate, mostly confined to the upper dermis, and composed of small lymphoid cells with minimal atypia (H&E, 100×). Scale bar = 200 μM. (**D**) In situ hybridization for Epstein–Barr virus–encoded small RNA (EBER) shows numerous positive cells in the skin (EBER, 100×). Scale bar = 200 μM.

**Figure 3 ijms-21-09314-f003:**
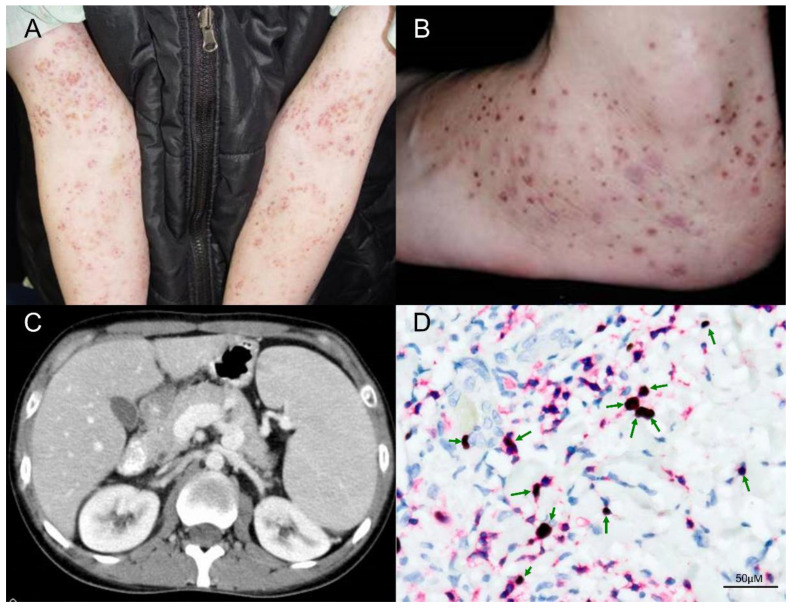
Clinical and pathologic features of a patient with HVLPD and development of HVLL. (**A**,**B**) The patient was a 31-year-old man who presented recurrent vesicles and papules on the face, hands and legs for 17 years. The cutaneous lesions extended to sun-protected areas, including upper arms and feet. (**C**) This patient had intermittent fevers, pancytopenia, and palpable neck lymph nodes, and the abdominal computerized tomography (CT) scan with contrast enhancement showed splenomegaly. (**D**) In the skin biopsy specimen, double staining of CD3 and EBER in situ hybridization proved EBV infection in many T cells (green arrows, 400×). Scale bar = 50 μM.

**Table 1 ijms-21-09314-t001:** A comparison of classic hydroa vacciniforme (HV), atypical or severe HV, and HV-like lymphoma.

Characteristics	Classic HV	Atypical/Severe HV	HVLL
**Race**	Whites and 40–50% of nonwhites	Asians and Latin Americans	Asians and Latin Americans
**Age**	Early childhood (1–7 years) and around or after puberty (12–16 years)	The median age at diagnosis is 8 years (range: 1–15 years)	Older than classic HV and atypical/severe HV
**Gender**	No gender bias	No gender bias	No gender bias
**Cutaneous involvement**	Sun-exposed areas	Sun-exposed and sun-protected areas	Sun-exposed and sun-protected areas
**Clinical presentation**	Papulovesicular lesions and crusted bullae on light-exposed areas	Similar to classic HV at the early stage, but extensively recur with progression to sun-protected areas	Long-term recurrent cutaneous lesions with progression to extracutaneous involvement and hematological abnormalities
**Systemic symptoms**	Absent	Occasional, especially after long-term recurrence	Present (e.g., fever, hepatitis, leukopenia, lymphadenopathy, hepatosplenomegaly, and hemophagocytic syndrome)
**Photoprovocation**	Usually positive	Often negative	Usually negative
**EBV DNA load (blood)**	Normal	Normal or mild increased	Highly increased
**Histopathology**	Reticulated epidermal degeneration and dense perivascular infiltration of mainly T cells	Similar to classic HV	Dense and extensive lymphoid infiltrate in soft tissue and extracutaneous organs. Lymphoid cells often show minimal or mild atypia.
**EBER in situ hybridization**	Positive, mainly in T-cells	Positive, mainly in T-cells	Positive, mainly in T-cells
**TCR gene rearrangement**	Polyclonal	Polyclonal or monoclonal	Monoclonal
**Genetic alteration**	No definitive finding	No definitive finding	GATA2 deficiency
**Treatment**	Sun protection	Sun protection, immunomodulators,	Immunomodulators, hematopoetic stem cell transplantation
**Prognosis**	Self-limited	Chronic indolent course with recurrent or refractory events. Unusual but may evolve into HVLL after dozens of years.	Poor and high risk of mortality

EBER: Epstein–Barr virus–encoded small RNA, TCR: T-cell receptor gene rearrangements.

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
