# Peer review of "Hydroa Vacciniforme and Hydroa Vacciniforme-Like Lymphoproliferative Disorder: A Spectrum of Disease Phenotypes Associated with Ultraviolet Irradiation and Chronic Epstein–Barr Virus Infection"

_ijms, 2020, doi:10.3390/ijms21239314_

Round 1
Reviewer 1 Report
This is a well written review dealing with a complex genodermatosis (Hydroa vacciniforme (HV).
I have no concern with this review but only a few minor points with regard to its presentation.
First, since clinical presentation of HV or related forms is rather complicated, in order to increase its attractiveness and wider readership, I would suggest authors to include a table summerazing HV (and derivative pathologies) presentations, causes, world distribution etc.
Second, figure legends only include magnification without scale bar.
In addition, immune cells infiltrations in patients skin could be better documented using specific markers as detected by indirect immuno histochemistry or fluorescence.
Author Response
Reviewer 1:
Thank you for your many suggestions regarding this manuscript. We have addressed all of your questions which are highlighted in red in the revised manuscript.
Q1 This is a well written review dealing with a complex genodermatosis (Hydroa vacciniforme (HV). I have no concern with this review but only a few minor points with regard to its presentation. First, since clinical presentation of HV or related forms is rather complicated, in order to increase its attractiveness and wider readership, I would suggest authors to include a table summarizing HV (and derivative pathologies) presentations, causes, world distribution etc.
- Thank you for your kind words and professional recommendation. Based on your suggestion, we added table 1 to summarize different phenotypes in our revised manuscript with their characteristics and comparison.
Q2. Second, figure legends only include magnification without scale bar. In addition, immune cells infiltrations in patients’ skin could be better documented using specific markers as detected by indirect immunohistochemistry or fluorescence.
- Thank you for this recommendation. We have added scale bars in all microscopic figures included in both the figures and figure legends. We agree that the immune cells infiltrations might be better documented using specific markers as detected by indirect immunohistochemistry or fluorescence. However, the skin biopsy specimens of both representative archived cases have been consumed with prior numerous immunohistochemistry (IHC) stains and serial T-cell clonality rearrangement (TCR) tests. In consequence, we cannot provide further stains of indirect immunohistochemistry or fluorescence. However, since IHC stains and TCR tests are relatively accessible and practical in most pathology laboratories, the figures might be more diagnostically useful.
Reviewer 2 Report
This is a interesing review manuscript to describe the disease and underlying mechanisms of Hydroa vacciniforme and hydroa vacciniforme-like lymphoproliferative disorders. However, the molecular mechanisms and immunological mechanisms need to be clearly discussed.
Major Comments:
- Figure 2a should be deleted. Due to the ethical issues, this 17 year old patient can be easly identified and tracked down. This is not ethical.
- The authors need to discuss by using thier own assessment on the molecular cellular and immunological mechanisms underlying the genetics under headline 5.
- The authors need to discuss by using thier own assessment on the molecular cellular and immunological mechanisms underlying the genetics under especially under headline 6. Molecular and genetic characteristics.
Author Response
Reviewer 2:
Thank you for your suggestions. According to your comments, we revised the relevant parts in the current manuscript. We have addressed the issues raised by you, and the amendments are highlighted in red in the revised manuscript.
This is a interesing review manuscript to describe the disease and underlying mechanisms of Hydroa vacciniforme and hydroa vacciniforme-like lymphoproliferative disorders. However, the molecular mechanisms and immunological mechanisms need to be clearly discussed.
Major Comments:
Q1. Figure 2a should be deleted. Due to the ethical issues, this 17 year old patient can be easily identified and tracked down. This is not ethical.
- Thank you for your kind words and professional recommendation. We have replaced figure 2a with the lesions of his earlobe. We believe the revised figures are not easily identified. Thank you for your precious time and expertise.
Q2. The authors need to discuss by using their own assessment on the molecular cellular and immunological mechanisms underlying the genetics under headline 5.
- We thank the reviewer for this suggestion. We have added some sentences to share our molecular experience using the T-cell receptor gene clonality to assess HVLPDs (headline 5, lines 271-277).
Q3. The authors need to discuss by using their own assessment on the molecular cellular and immunological mechanisms underlying the genetics under especially under headline 6. Molecular and genetic characteristics.
- Thank you for this suggestion. Because of the rarity and limits of molecular targets reported, very few molecular and genetic characteristics have been reported, even using whole genome sequencing. Moreover, although we have reported several cases of HVLPDs, we have currently no data about the genetic features of HVLPDs. Therefore, we regret that we are unable to share our experience on genetic assessment further. We have addressed the comments in headline 6 (lines 281-282).
Round 2
Reviewer 2 Report
The Authors have revised the manuscript by following reviewer's suggestions. It is a nice work.